Deep vision-based real-time hand gesture recognition: a review

Cui Cui 1 2
http://orcid.org/0000-0002-0244-1622 Sunar Mohd Shahrizal 1 2 shahrizal@utm.my
Eg Su Goh 1 2
1 Media and Game Innovation Centre of Excellence (MaGICX), Institute of Human Centered Engineering (iHumEn), Universiti Teknologi Malaysia , Johor Bahru , Malaysia
2 Faculty of Computing, Universiti Teknologi Malaysia , Johor Bahru , Malaysia
Comai Sara
Electronic publication date: 2025 Jun 24
Publication date: 2025
Volume: 11
Electronic Location ID: e2921
Received 2024 Dec 19; Accepted 2025 May 6
Copyright: © 2025 Cui et al.
Copyright year: 2025
Copyright holder: Cui et al.
License: This is an open access article distributed under the terms of the Creative Commons Attribution License, which permits unrestricted use, distribution, reproduction and adaptation in any medium and for any purpose provided that it is properly attributed. For attribution, the original author(s), title, publication source (PeerJ Computer Science) and either DOI or URL of the article must be cited.
License URL: https://creativecommons.org/licenses/by/4.0/

Keywords: Deep learning, Evaluation metric, Self-created datasets, Underlying models, Ablation study

Funding: Ministry of Higher Education Malaysia FRGS/1/2022/ICT10/UTM/02/1 This work was supported by the Ministry of Higher Education Malaysia under Fundamental Research GrantScheme [FRGS/1/2022/ICT10/UTM/02/1]. The funders had no role in study design, data collection and analysis, decision to publish, or preparation of the manuscript.

==============================
Hand gesture recognition is an approach to comprehending human body language, applied in various fields such as human-computer interaction. However, some issues remain in edge blurring generated by complex backgrounds, rotation inaccuracy induced by fast movement, and delay caused by computing cost. Recently, the emergence of deep learning has ameliorated these issues, convolution neural network (CNN) enhanced edge clarity, long-short term memory (LSTM) improved rotation accuracy, and attention mechanism optimized response time. In this context, this review starts with the deep learning models, specifically CNN, LSTM, and attention mechanisms, which are compared and discussed from the utilization rate of each, their contribution to improving accuracy or efficiency, and their role in the recognition stage, like feature extraction. Furthermore, to evaluate the performance of these deep learning models, the evaluation metrics, datasets, and ablation studies are analyzed and discussed. The choice of evaluation metrics and dataset is critical since different tasks require different evaluation parameters, and the model learns more patterns and features from diverse data. Therefore, the evaluation metrics are categorized into accuracy and efficiency. The datasets are analyzed from self-created to public datasets. The ablation study is summarized in four aspects: similar underlying models, integrating specific models, pre-processing, others. Finally, the existing research gaps and further research on accuracy, efficiency, application range, and environmental adaptation are discussed.

Introduction

In the ever-evolving landscape of technology and human-computer interaction, hand gesture recognition (HGR) emerges as a transformative force. It bridges the divide between human communication and the digital world, impacting diverse domains such as augmented reality (AR), virtual reality (VR), and sign language interpretation.

HGR accomplishes communication between humans and machines by tracking hand gestures and recognizing their representation, converting them into semantically meaningful commands. Based on data sources and technologies used for capturing and analyzing gestures, HGR can be classified into vision-based HGR, sensor-based HGR, wearable-based HGR, etc. Among them, vision-based HGR is primarily concerned with capturing signals through the camera, calculating the positional information of the hand through models, evaluating the gesture posture, and then processing it into comprehensible information. Vision-based real-time HGR further emphasizes efficiency and immediate responsiveness in simultaneously capturing gestures and executing real-time data analysis, ensuring fast response in dynamic interaction scenes. Given this, vision-based dynamic HGR has been further advanced, becoming a hot research topic in recent years in several application scenarios (Zhang, Wang & Lan, 2020), such as robotics (Nogales & Benalcázar, 2021), military (Naik et al., 2022), clinical operations (Lanza et al., 2023), risk warning (Wang et al., 2023b), agriculture (Moysiadis et al., 2022), remote collaboration (Tian et al., 2023), AR (Yusof et al., 2016), entertainment (Cui, Sunar & Su, 2024) and education (Su et al., 2022). In robotics control, it empower robots flexibility and precision; in the military, it supports real-time silent communication and command; in clinical operations, it supports surgeons to control surgical equipment in real time; in risk warning, it realizes instant safety response through gesture signals; in agriculture, it regulates drones in real time through gestures for crop monitoring or pesticide spraying; and in the fields of remote collaboration, AR and education, it enhances the naturalness and interactivity of the user experience.

Two momentous metrics for vision-based real-time HGR are instant response and recognition accuracy. However, challenges persist in accuracy and efficiency, such as inaccuracy in rotation or fast movement, high computing costs in video processing, and long response times. Recently, the emergence of deep learning has enabled some improvement in these issues. Since 2018, deep learning models have been widely adopted by an increasing cohort of HGR researchers. Many researchers have successfully implemented deep learning models to enhance the performance of HGR, such as structured dynamic time warping (SDTW) (Tang et al., 2018), InteractionFusion (Zhang et al., 2019), and a two-branch fusion deformable network (Liu & Liu, 2023). The previously mentioned deep learning models are deeply involved in each key stage of HGR, which plays an essential role in promoting system performance and enhancing the interaction experience. The key step of HGR involves acquiring data, data pre-processing, feature extraction, hand segmentation, hand detection and tracking, and classification, as shown in Fig. 1. The sensor captures continuous hand gestures. These data are cleaned through a pre-processing stage to remove invalid frames. Meanwhile, the hand is extracted from the intricate background to obtain a segmented hand area. Simultaneously, hand location is detected within the frame while tracking the continuous location, rotation, and orientation. At the core of the process is feature extraction, such as skin color, skeleton, and spatiotemporal information. Finally, the identified hand gestures are classified.

Figure 1 Pipeline of hand gesture recognition.

In this context, this review embarks on a comprehensive exploration of vision-based deep learning methods and evaluation techniques in real-time HGR, spanning from 2018 to 2024. Our analysis focuses on three core aspects: the model, model evaluation, and research gaps. Although many researchers have summarized the literature in this area regarding methods, applications, challenges, and so on in Table 1, they have not detailed the relationship between improvements and models, or the influence of key stages of HGR on the models within these methods. Unlike previous reviews or surveys, this review summarizes and discusses these aspects, as the strengths and weaknesses of the model at each stage directly affect the final recognition. Furthermore, we elaborated on and analyzed model performance across the dataset to highlight the intrinsic relevance of model performance concerning both models and datasets.

Table 1 The previous related reviews or surveys from 2018 to 2024.

				Elements covered in the study	
Reference	Style	Count	Timeline	Acquisition device	Methods	Evaluation metric	Datasets	Application	Challenge	Future direction	
Xia et al. (2019)	Survey	N/A	N/A	✓	✓	–	–	–	–	✓	
Zulpukharkyzy Zholshiyeva et al. (2021)	Survey	70	2012–2021	–	✓	–	–	✓	–	–	
Kaur & Bansal (2022)	Survey	19	N/A	–	✓	✓	–	✓	✓	✓	
Al-Shamayleh et al. (2018)	Review	100	N/A	✓	✓	–	–	✓	✓	✓	
Yasen & Jusoh (2019)	Review	111	2016–2018	–	✓	–	–	✓	✓	–	
Vuletic et al. (2019)	Review	148	1980–2018	✓	✓	–	–	✓	–	✓	
Nogales & Benalcázar (2021)	Review	40	2015–2019	✓	✓	✓	✓	–	✓	–	
Chakraborty et al. (2018)	Review	N/A	N/A	✓	✓	–	–	–	✓	–	
Oudah, Al-Naji & Chahl (2020)	Review	N/A	N/A	–	✓	–	–	✓	✓	–	
Mohamed, Mustafa & Jomhari (2021)	Review	98	2014–2020	✓	✓	✓	✓	–	–	✓	
Sarma & Bhuyan (2021)	Review	N/A	N/A	–	✓	–	✓	–	✓	✓	
Sha et al. (2020)	Review	N/A	N/A	–	✓	✓	✓	–	–	✓	
Al Farid et al. (2022)	Review	108	2012–2022	–	✓	–	–	–	✓	✓	

Based on the 2012 Association for Computing Machinery (ACM) computing classification, the focus area of this review is highlighted in blue color as depicted in Fig. 2. The goal of this review is to review previous research on vision-based real-time HGR by deep learning. The main contribution of this review is summarized as follows: To summarize and discuss the last 7 years of real-time hand gesture recognition methods using deep learning. Linking the deep learning models with improvement and role in key stage.

To analyze and discuss the evaluation metrics and datasets for evaluating, training, and testing the model, especially the comparison between self-created datasets and public datasets, and the technical evaluation.

To analyze the relationship between the ablation study and the selection of underlying models

To categorize the existing research gaps and suggest potential directions.

Figure 2 ACM computing classification system.

The blue color is the research direction to which the research content of this review subsumes.

Related work

Since the application of real-time HGR has been promoted, research efforts in this field are on a constant upward trajectory. To facilitate the research of subsequent researchers, many researchers have summarized and analyzed the related literature. Some of them are more comprehensive in their analytical summaries, including static and dynamic HGR, vision-based and wearable-based HGR (Nogales & Benalcázar, 2021; Tang et al., 2018; Zhang et al., 2019; Liu & Liu, 2023). Others have some review and survey on vision-based HGR. These related articles are listed in Table 1.

Acquisition device

Some of these reviews and surveys summarize and generalized vision-based real-time HGR from diverse perspectives. The analyses and comparisons were carried out in the following aspects: analyzing single cameras, active techniques, and invasive techniques (Xiong et al., 2021); collecting in cameras, sensors, and wearable devices (Vuletic et al., 2019); comparing the difference between Sensor devices and vision-based devices (Vuletic et al., 2019; Al-Shamayleh et al., 2018; Sarma & Bhuyan, 2021); summarizing basic information and defects of 2D camera or a 3D camera (Chakraborty et al., 2018); comparing sensor technologies (Xia et al., 2019). Among them, the surface electromyography (sEMG) sensors (Yasen & Jusoh, 2019) and leap motion controller were generic acquisition tools (Nogales & Benalcázar, 2021; Mohamed, Mustafa & Jomhari, 2021).

Method

For methods or techniques, Al-Shamayleh et al. (2018) discovered that most studies focus on appearance-based HGR of vision-based HGR. Some reviews or surveys analyzed hand detection, segmentation, and classification technologies (Vuletic et al., 2019; Sarma & Bhuyan, 2021; Xia et al., 2019; Al Farid et al., 2022; Oudah, Al-Naji & Chahl, 2020; Zulpukharkyzy Zholshiyeva et al., 2021). Feature extraction is automatically obtained using deep learning such as convolutional neural networks (CNN) or long short-term memory (LSTM) (Nogales & Benalcázar, 2021; Sha et al., 2020). Moreover, the classification models included k-nearest neighbors (KNN), dynamic time warping (DTW), support vector machine (SVM), artificial neural network (ANN), LSTM (Nogales & Benalcázar, 2021). Yasen & Jusoh (2019) found that an ANN is a widely used classifier. Chakraborty et al. (2018) summarized advantages and disadvantages of various classifiers.

Application

For application, the HGR is applied in various areas such as robot control (Vuletic et al., 2019; Oudah, Al-Naji & Chahl, 2020; Zulpukharkyzy Zholshiyeva et al., 2021), sign language recognition (Vuletic et al., 2019; Al-Shamayleh et al., 2018; Sarma & Bhuyan, 2021; Yasen & Jusoh, 2019; Oudah, Al-Naji & Chahl, 2020; Zulpukharkyzy Zholshiyeva et al., 2021), healthcare (Vuletic et al., 2019; Sarma & Bhuyan, 2021; Oudah, Al-Naji & Chahl, 2020; Zulpukharkyzy Zholshiyeva et al., 2021), entertainment (Vuletic et al., 2019; Al-Shamayleh et al., 2018; Oudah, Al-Naji & Chahl, 2020), the most sign language recognition among them (Al-Shamayleh et al., 2018; Yasen & Jusoh, 2019).

Evaluation metric

For evaluation metrics, some reviews or surveys analyzed many aspects, such as parameters or tools of accuracy (Kaur & Bansal, 2022), processing time (Nogales & Benalcázar, 2021), and recognition accuracy (Nogales & Benalcázar, 2021; Mohamed, Mustafa & Jomhari, 2021; Sha et al., 2020).

Datasets

The datasets are created by Kinect, Leap Motion, Intel RealSense, or an Interactive gesture camera (Nogales & Benalcázar, 2021). The datasets are sorted by fingerspelling, isolated and continuous gestures (Mohamed, Mustafa & Jomhari, 2021). Sarma & Bhuyan (2021) listed the content and links of some datasets. Sha et al. (2020) described the details of seven isolated gesture datasets.

Research gap and challenge

The majority of reviews and surveys identify potential research directions derived from analyzing research gaps or challenges.

The challenge of vision-based real time HGR includes system (Al-Shamayleh et al., 2018), complex background (Al-Shamayleh et al., 2018; Chakraborty et al., 2018; Yasen & Jusoh, 2019; Oudah, Al-Naji & Chahl, 2020; Kaur & Bansal, 2022), illumination variation (Chakraborty et al., 2018; Oudah, Al-Naji & Chahl, 2020; Kaur & Bansal, 2022), gesture-related challenges (Al-Shamayleh et al., 2018; Chakraborty et al., 2018; Kaur & Bansal, 2022), highly accuracy and efficient recognition (Nogales & Benalcázar, 2021; Sarma & Bhuyan, 2021; Chakraborty et al., 2018; Kaur & Bansal, 2022), reasonableness of the technology and its application (Vuletic et al., 2019), overfitting in the datasets (Yasen & Jusoh, 2019), matching issues in datasets (Oudah, Al-Naji & Chahl, 2020).

Future research direction

For future scope, Al-Shamayleh et al. (2018) introduced potential research directions including syntactic interpretation, hybrid methods, smartphone sensors, normalization, and real-life systems. Kaur & Bansal (2022) proposed potential research directions including dynamic hand classification methods in videos, number of classifications, recognition complexity and time reduction hand gesture detection, tracking and classification in complex backgrounds, real-time requirement, and user experience improvement (Xia et al., 2019).

The combination of hand gestures and speech is a research direction warranting attention in the future (Vuletic et al., 2019). Increased collective volume of datasets, feature integration, and reducing computational cost will be the future directions (Mohamed, Mustafa & Jomhari, 2021). Gesture communication will instead involve multi-step interactions (Sarma & Bhuyan, 2021). Research activities may advance in large-scale datasets, temporal or spatial relations, the gap between virtual and real scenes (Sha et al., 2020).

The preceding reviews and surveys have not analyzed the relationship between improvement and models, despite many of them organizing applications of models. Meanwhile, they have not analyzed the functions of the model in each stage of HGR. In addition, most of them focus on obtaining data and public datasets, rarely summarizing self-created datasets.

This review, by contrast, summarizes and discusses the roles of models at each stage of the recognition process and their impact on overall performance. Among the highlights are the last 7 years studies in vision-based real-time HGR deep learning models including CNN, LSTM, attention mechanisms, and others.

In addition, this review organizes the evaluation metrics from both accuracy and efficiency perspectives to inform future researchers in the selection of evaluation parameters. Furthermore, analyzing the distribution of self-created datasets and public datasets provides a reference for choosing the datasets for training and testing the models.

Methodology

In this section, we are based on the understanding of the definition of Bjørnson & Dingsøyr (2008) and incorporate the review methodology detailed by Kitchenham et al. (2009). Completing our study according to Lavallée, Robillard & Mirsalari (2013) and the PRISMA statement (Page et al., 2020), the research methodology for literature selection includes six steps: planning, research questions, search strategy, inclusion and exclusion criteria, quality assessment, data extraction. PRISMA is universally acknowledged as the crucial framework for writing reviews. It provides a comprehensive guideline for researchers to complete reviews in a structured, methodological manner. Therefore, we follow this framework to finish this section. The selection process of the studies consists of two main stages, as shown in Fig. 3, searching for and analyzing previous literature.

Figure 3 The process of selecting studies consists of two parts, previous literature search and analysis.

The previous literature search contains three steps, searching in the literature repository, duplicate removal, and adding from others.

Research questions

The research question is essential to guide the overall review process. We define the following three research questions (RQ), aiming to organize the results relevant to the models, performance evaluation, and research gaps. These include the improvement and limitation of methods, the relationship between models and improvement, the models in key stages of HGR processing, the evaluation metrics and datasets, and the research gaps and future directions. All of them focus on methods for real-time HGR using deep learning (DL). RQ1. What type of models and procedures are used in the real time HGR through deep learning?

RQ2. What are the performance metrics used to evaluate the HGR models?

RQ3. Which research gap remains in real time HGR using deep learning?

The findings of each RQs are analysis and discussed in sections of RQ1, RQ2, and RQ3. The overall findings organized in Synthesis of Findings.

Search strategy

This section involves designing and implementing a structured literature search process to locate inclusively relevant studies by using keywords and databases. The period of search for references is from January 2018 to May 2024. References to the sources used in the literature repositories: IEEE Xplore, Web of Science, Scopus and Springer. To ensure covering relevant literature, keywords, and synonymous words related to RQs are selected, as demonstrated in Table 2. The number of studies for each literature repository is listed in Table 3.

Table 2 The keywords for searching studies.

ID	Keywords	
K1	‘Vision based’ and ‘Real Time’ and ‘Hand Gesture Recognition’ and ‘Deep Learning’	
K2	‘Vision based’ and ‘Dynamic’ and ‘Hand Gesture Recognition’ and ‘Deep Learning’	
K3	‘Vision based’ and ‘Real Time’ and ‘Hand Gesture Recognition’ and ‘RNN’	
K4	‘Vision based’ and ‘Real Time’ and ‘Hand Gesture Recognition’ and ‘CNN’	
K5	‘Vision based’ and ‘Real Time’ and ‘Hand Gesture Recognition’ and ‘LSTM’	
K6	‘Vision based’ and ‘Real Time’ and ‘Hand Gesture Recognition’ and ‘Attention Mechanism’	

Table 3 The number of the studies for each literature repository and keywords.

Literature repositories	Keywords	Total	
K1	K2	K3	K4	K5	K6		
IEEE Xplore	59	29	2	32	6	29	157	
ACM digital library	131	112	35	105	58	7	448	
Scopus	64	62	3	48	9	2	188	
Springer	105	71	25	70	28	36	335	
Web of science	39	34	1	27	6	3	110	
Science direct	35	30	11	31	15	15	137	

In Fig. 3, the initial search produced 1,375 studies that were filtered for duplicate removal, and 1,021 studies were chosen. In addition, 32 studies were collected from Google Scholar, CVPR, ICCV, ECCV, and other repositories and conference articles to ensure sufficient relevant literature was searched. In the first stage, 1,053 studies were conducted. In the second stage, 199 studies were extracted by inclusion and exclusion criteria, and then forty-seven studies were finally selected by quality assessment.

Inclusion and exclusion criteria

To determine whether a primary study would help answer the RQs and to ensure the completeness and accuracy of the search strategy. To ensure the selected articles can cover all RQs, the inclusion and exclusion criteria were created in Table 4.

Table 4 Inclusion and exclusion criteria.

Inclusion criteria	The model of real-time or dynamic Hand Gesture Recognition (HGR)	
3D-based, Vision-based and using deep learning	
Exclusion criteria	No indication that the model is real-time or dynamic HGR	
No vision-based and no using Deep Learning	
Gesture recognition without human hands	

Quality assessment

The quality of the references was assessed by answering the following seven quiz questions, each with three responses, and their scores: “Yes” = 1, “Partly” = 0.5, and “No” = 0. The results shown in Table 5, the designed questions facilitated answering the RQs mentioned previously, followed by summing and ranking the scores. Were the research purposes of the study clear?

Was the structure of the HGR model shown?

Was the results of experiments shown?

Were the contributions of the study clear?

Did the article mention the future works?

Was the limitation explicitly mentioned?

Was the article published in an accreditation source?

Table 5 Quality assessment criteria.

No.	Quality assessment questions	Criteria scores	
QAC1	Were the research purposes of the study clear?	“Yes” = 1/“No” = 0	
QAC2	Was the structure of the HGR model shown?	“Yes” = 1, “Partly” = 0.5, and “No” = 0.	
QAC3	Was the results of experiments shown?	“Yes” = 1, “Partly” = 0.5, and “No” = 0.	
QAC4	Were the contributions of the study clear?	“Yes” = 1, “Partly” = 0.5, and “No” = 0.	
QAC5	Did the article mention the future works?	“Yes” = 1/“No” = 0	
QAC6	Was the limitation explicitly mentioned?	“Yes” = 1/“No” = 0	
QAC7	Was the article published in an accreditation source?	Rank by IF/JCR Q1 = 2 or CVPR, ICCV, ECCV = 2, Rank by IF/JCR Q2 = 1.5, Rank by IF/JCR Q3 or Q4 = 1, no ranking = 0	
Note:

*QAC, Quality Assessment Criteria.

Data extraction

In the second stage of Fig. 3, the 199 studies were selected by title, abstract, and full article according to inclusion and exclusion criteria. Moreover, the studies were assessed based on quality assessment, the scores presented in Table 6. Finally, forty-seven studies were analyzed in depth for the three RQs.

Table 6 Data extracted results.

Ref.No.	Reference	Scores	
R1	Tang et al. (2018)	8	
R2	Zhang et al. (2019)	8	
R3	Zhang, Wang & Lan (2020)	8	
R4	Tang et al. (2021)	8	
R5	Sharma & Singh (2021)	8	
R6	Liu & Liu (2023)	8	
R7	Shanmugam & Narayanan (2024)	8	
R8	Rastgoo et al. (2024)	8	
R9	Balaji & Prusty (2024)	7.5	
R10	Zhang, Tian & Zhou (2018)	7.5	
R11	Fang et al. (2019)	7.5	
R12	Li et al. (2019)	7.5	
R13	Lu et al. (2019)	7.5	
R14	Ozcan & Basturk (2019)	7.5	
R15	dos Santos, Samatelo & Vassallo (2020)	7.5	
R16	Rahim, Shin & Islam (2020)	7.5	
R17	Tellaeche Iglesias et al. (2021)	7.5	
R18	Wang et al. (2023a)	7.5	
R19	Ng et al. (2022)	7.5	
R20	Cao, Li & Shin (2022)	7.5	
R21	Jain, Karsh & Barbhuiya (2022)	7.5	
R22	Rajalakshmi et al. (2023)	7	
R23	Lu et al. (2024)	7	
R24	Huang et al. (2023)	7	
R25	Li et al. (2018)	7	
R26	Patil & Subbaraman (2019)	7	
R27	Ameur, Khalifa & Bouhlel (2020)	7	
R28	Jiang et al. (2021)	7	
R29	Rubin Bose & Sathiesh Kumar (2021)	7	
R30	Li et al. (2021b)	7	
R31	Yu et al. (2021)	7	
R32	Bose & Kumar (2022)	7	
R33	Chen et al. (2023)	6.5	
R34	Xiao et al. (2023)	6.5	
R35	Hou et al. (2023)	6.5	
R36	Liu et al. (2019)	6.5	
R37	Verma & Choudhary (2020)	6.5	
R38	Do et al. (2020)	6.5	
R39	Li et al. (2021a)	6.5	
R40	Taranta et al. (2021)	6.5	
R41	Verma (2022)	6.5	
R42	Wang (2022)	6.5	
R43	Yadav et al. (2022)	6.5	
R44	Dubey (2023)	6.5	
R45	Guler & Yucedag (2022)	6.5	
R46	Haroon et al. (2022)	6.5	
R47	Rastgoo, Kiani & Escalera (2022)	6.5	

Biased evaluation criteria

This section analyzes the following constraints on this review and alleviation options.

First, searching for articles is incomplete. The proximity of the number of searched articles to the total related literature in the literature database determines the completeness of the search articles. That is directly associated with the selection of the database and the definition of searching keywords or their synonyms. To address these issues, we conducted the search strategy in the earlier section.

Second, inclusion and exclusion criteria may not apply to the three RQs. To obviate this, the formal inclusion and exclusion criteria are formulated in the previous section.

Third, quality assessment is not objective. To avoid these issues, this review developed eight questions to reduce this trend.

Lastly, data extraction is inaccurate. To solve this problem, data are extracted by a comprehensive method on the basis of the three RQs, inclusion and exclusion criteria, and quality assessment.

RQ1: what type of models and procedures are used in the real time HGR through deep learning?

In response to RQ1, this section provided a detailed statistical summary and in-depth analysis, concentrating on the underlying models in methods, the HGR models specifically designed for key stages in real-time recognition, and analysis of their strengths and weaknesses. These statistics not only reveal the advancement of real-time HGR technologies but also serve as essential references for researchers in selecting optimal models for specific application scenarios.

The statistics shown below are collected from forty-seven articles by reading them. The data on methods, model composition, methods employed in key stages of HGR, advantages and disadvantages are listed in a spreadsheet, generated by professional graph and chart generating software.

Underlying models in methods

Since each deep learning model has advantages and disadvantages, methods designed with different underlying models imply obtaining different research results. Hence, the option of models is directly related to overall performance. Figure 4, Table 7 demonstrate the trends and distribution of underlying models in methods.

Figure 4 Trends in the use of CNN, LSTM, attention mechanism, multiple model combinations, and other deep learning models from 2018 to 2024.

Table 7 Underlying models in methods.

Ref. No.	Methods	CNN	LSTM	Attention mechanism	Others	
R1	Structured Dynamic Time Warping (SDTW)	–	✓	–	✓	
R2	InteractionFusion	–	✓	–	✓	
R3	Short-Term Sampling Neural Networks (STSNN)	✓	✓	–	–	
R4	Selective Spatiotemporal Features Learning (SeST).	✓	✓	–	–	
R5	Gesture-CNN (G-CNN)	✓	–	–	–	
R6	A Two-branch Fusion Deformable Network with Gram Matching	✓	✓	–	✓	
R7	Modified Deep Convolutional Neural Network-based Hybrid Arithmetic Hunger Games (MDCNN-HAHG)	✓	–	–	–	
R8	A Transformer-based with a C3D; AutoEncoder (AE) on LSTM network	✓	✓	✓	–	
R9	Multimodal Fusion Hierarchical Self-attention Network (MF-HAN)	–	–	✓	–	
R10	HandSense	✓	–	–	–	
R11	An Integrated Framework Based on the Covariance Matrix	✓	–	✓	–	
R12	A Spatiotemporal Attention-based ResC3D model	✓	–	–	–	
R13	A Lightweight Inflated 3D ConvNets (I3D)	✓	–	–	–	
R14	ABC Tuned-CNN structure	✓	–	–	–	
R15	Star RGB and Dynamic Gesture Classifier	✓	–	–	–	
R16	A non-touch character writing system	✓	–	–	–	
R17	The optimized Darknet CNN architecture	✓	–	✓	–	
R18	A Two-branch Hand Gesture Recognition Approach (HGRA)	✓	–	✓	–	
R19	AA-A2J and AA-3DA2J	✓	–	✓	–	
R20	A Transformer-based Network	✓	–	✓	–	
R21	The Encoded Motion Image (EMI)	–	✓	–	✓	
R22	The hDNN-SLR Framework	–	–	–	✓	
R23	3D DenseNet-BiLSTM	✓	✓	–	–	
R24	Deep Robust Hand Gesture Network (RGRNet)	✓	–	–	✓	
R25	Fisher Bidirectional Long-Short Term Memory (F-BiLSTM) and Fisher Bidirectional Gated Recurrent Unit (F-BiGRU)	✓	✓	–	–	
R26	Hough Transform (HT) and Artificial Neural Network (ANN)	✓	✓	✓	–	
R27	Hybrid Bidirectional Unidirectional LSTM (HBU-LSTM)	✓	✓	–	–	
R28	The Generative Adversarial networks (GAN) Model and the Mask R-CNN	✓	✓	✓	–	
R29	The Optimum Deep Residual Network (RetinaNet-DSC)	✓	–	–	–	
R30	A Lightweight 3D Inception-ResNet	✓	–	–	✓	
R31	Resnet-101 and SSD with attention mechanism (TA-RSSD) and Temporal Attentional LSTM (TA-LSTM)	✓	✓	✓	✓	
R32	The Deep Single-stage CNN Model (Hybrid-SSR)	✓	–	–	–	
R33	The RPCNet Module	✓	–	✓	–	
R34	TSM-ResNet50	✓	✓	–	–	
R35	Kernel Optimize Accumulation (KOA), Union Frame Difference (UDF), and CLSTM	–	✓	–	✓	
R36	The Radial Basis Function (RBF) Neural Networks	–	–	–	✓	
R37	A Novel Grassmann Manifold Based Framework	✓	–	–	–	
R38	A Multi-level Feature LSTM with Conv1D, the Conv2D Pyramid, and the LSTM Block	–	✓	–	✓	
R39	3D-Ghost and Spatial Attention Inflated 3D ConvNet (3DGSAI)	✓	–	✓	–	
R40	Machete	–	–	–	✓	
R41	A Two-Stream Hybrid Model (CNN + BGRU model)	✓	✓	–	✓	
R42	The Normalized 2D SDD Features and A Priori Knowledge	✓	–	–	–	
R43	A Pixel-wise Semantic Segmentation (SegNet) Model with VGG16; SegNet network, point tracker and Kalman filter (SDT); a Deep Convolutional Neural Network (DCNN)	–	✓	✓	✓	
R44	The Adaptive Region-based Active Contour (ARAC), the Principal Component Analysis (PCA), the Optimized Probabilistic Neural Network (PNN), the Opposition Strategic Velocity Updated Beetle Swarm Optimization (OSV-BSO)	✓	✓	–	–	
R45	Convolutional Capsule Neural Network (CCNN) Model	✓	✓	–	–	
R46	A Scale-Invariant Feature Transfor (SIFT)	✓	–	✓	–	
R47	Single Shot Detector (SSD), CNN, LSTM, Discriminative Hand-related features (SVD)	✓	✓	–	–	

Trends in underlying model utilization

Analyzing these forty-seven articles, as demonstrated in Fig. 4, many researchers are incorporating deep learning models for real-time HGR. The trend is to peak in 2022.

Underlying models

Table 7 lists the underlying deep-learning models implemented in each method, with some methods having just one and others combining many models.

The HGR models for key stage in real-time HGR

The final performance of the HGR will be affected by the selection, alteration, or improvement of the models in the key stages of the HGR process. These critical steps include pre-processing data, feature extraction, hand detection, hand segmentation, hand classification, and hand tracking. In Fig. 5, the models that concentrate on research to improve feature extraction are the largest, with minimal differences in the number of other stages, as follows.

Figure 5 Compared the models in each key stage, absolutely, feature extraction is the biggest one.

In particular, pre-processing plays a critical role in bridging the gap between data collection and model training while enhancing the accuracy and robustness of deep learning models. Despite its significance, pre-processing is rarely applied in the analyzed literature, as shown in Table 8.

Table 8 Pre-processing technologies.

Ref. No.	Technologies	
R3	Zoom out images; Rotation and crop	
R5, R29	Image resize and data labeling	
R6	A time-synchronized preprocessing method based on the RGB morphology and radar spectrum differences	
R9	Converted to tokens using a Pixel Encoder	
R12	Hybrid median filter	
R15	Star RGB	
R17	Transfer learning	
R25	Suppress noise (i.e., data smoothing) using Average Filter, Median Filter, and Butterworth Filter	
R28	Generative adversarial networks (GAN)	
R35	Compress input video stream	
R43	Data augmentation: translation and noise injection	
R44	Median filtering	
R46	Luminosity method based on gray-scale conversion of the input image	

Improvements and limitations in the HGR models

The researchers designed models with DL to solve the problems that still exist in real-time hand gesture recognition. Most of these models specialize in enhancing accuracy, efficiency, robustness, occlusion issues, and reducing consumption problems. Whereas some progress has been achieved, some limitations remain such as, partial accuracy, efficiency, robustness, application range. Figure 6 shows the contribution of model’s implementation to the whole performance enhancement for real-time HGR. The deeply detailed comparison are analysis in section ‘Comparative analysis of technologies in methods’.

Figure 6 Compares the relationship between the underlying models and the direction of improvement.

Most of the models contributed recognition accuracy improvements and efficiency advancements.

By the analysis, these methods improve one or more of the HGR’s performances with one model or combination. In Fig. 6, most of the researchers focus on enhancing accuracy in which CNN is used most than others.

Comparative analysis of technologies in methods

CNN and model combinations are the most widespread methods, as shown in Fig. 4 and Table 7. This is because each model has its merits and demerits; combining the underlying models can achieve higher accuracy and efficiency in HGR. Furthermore, most studies tend to enhance accuracy, as illustrated in Fig. 6. Additionally, Fig. 5 reveals that the majority of model research focuses on improving feature extraction. The pre-processing technologies contribute to enhancing model accuracy; fewer studies utilize this technique, as shown in the Fig. 5.

Underlying models in methods

Since CNN has a unique advantage in feature extraction to enhance accuracy, it is frequently used in methods, either by itself or combined with other models. The most prevalent of them are 2 deep convolutional neural network (DCNN) and 3DCNN. The 2DCNN is only for a single image, while the 3DCNN is for continuous frame images. 2DCNN can efficiently process 2D static images and also the video segmented into a sequence of static images by frame; however, most of the background information of video images is redundant, leading to inefficiency. Therefore, the 2DCNN-based models are constrained in modeling temporal relationships, whereas the 3DCNN-based models better capture temporal features at a computational cost.

In addition, numerous models are designed based on influential CNN architectures to optimize performance. Such as, AlexNet (Krizhevsky, Sutskever & Hinton, 2017), VGG16, VGG19 (Simonyan & Zisserman, 2014), GoogleNet (Szegedy et al., 2015), and ResNet (He et al., 2016), C3D (Tran et al., 2015), ResC3D (Tran et al., 2017), DenseNet (Huang et al., 2017).

Although CNN extracts features of a single image more accurately, front-to-back temporal dependencies remain insufficiently captured. Moreover, 3DCNN only extracts short-term features. On the contrary, LSTM is designed to associate previous information with the current task, with an additional forget gate compared to RNN, which determines the retention of earlier information at the current moment.

However, LSTM has a relatively complex model structure that is more time-consuming to train than CNN, and has a disadvantage in parallel processing. In contrast, the attention mechanism enables lightweight and parallel processing. It can filter out task-independent information while enhancing task-related information, but the results are not accurate enough.

Hence, some researchers integrate CNN or LSTM or attention mechanism or other deep learning models to obtain better performance. The advantages and disadvantages of these models are summarized in Fig. 7.

Figure 7 The advantages and disadvantages of CNN, LSTM, and attention mechanism and their integration in a real-time HGR pipeline.

The HGR models for key stage in real-time HGR

The key stage model is essentially designed to achieve better performance in real-time gesture recognition. In this case, improving the feature extraction model is significant, especially spatiotemporal feature extraction. The others are similar, as shown in Fig. 5. The reason for this is that the precision of feature extraction has a direct influence on not only the segmentation, tracking, and classification but also the precision and responsiveness of real-time HGR.

Moreover, in the recent seven years, pre-processing techniques have been employed to tackle specific challenges in image and video data. Pre-processing data decreases the amount of unnecessary data in training and testing, thereby enhancing accuracy and reducing computational consumption.

Various pre-processing techniques are applied to address inaccuracy. Image resizing (R5, R29) standardizes input data, such as resizing images to 300 × 300 pixels, ensuring consistency across datasets and facilitating model training. The hands are labeled by LabelImg (R29) to obtain accurate annotations for supervised learning. Rotation, cropping, and zooming (R3) further augment the dataset by simulating different perspectives, reducing overfitting, and improving the model’s generalization.

Noise reduction is another critical aspect to enhance accuracy, as noisy data can degrade model performance. Multiple filtering techniques are employed to suppress noise, including average filters, median filters, and Butterworth filters (R25), hybrid median filters (R12) and median filtering (R44). The luminosity method (R46) converts images to grayscale while retaining essential hand edges. What’s more, Transfer learning (R17) leverages pre-trained models to improve efficiency and accuracy for tasks with limited data. Translation and noise injection (R43) and generative adversarial networks (GANs) (R28) expand data diversity to enhance model robustness.

In particular, some pre-processing technologies for a special time-synchronized method (R6) leverage differences in RGB morphology and radar spectrum. Pixel tokenization via self-attention mechanisms (R9) is used to convert tokens to capture spatial dependencies. Additionally, compressing input video streams (R35) ensures lower computational cost.

Furthermore, in the recent 7 years, researchers have begun to refine hand segmentation models for deep learning and enhance the essential stage in alignment with application scenarios. This is because accurate segmentation is beneficial for precisely identifying hand gestures. In some cases, the vital stage of improvement links to the application context. At the classification stage, for example, enhancement models are chiefly applied in sign language recognition and handwriting trajectory recognition.

Improvements and limitations in the HGR models

Researchers have undertaken to enhance the accuracy and efficiency, whereas these two aspects are still primary issues to be addressed, as shown in Fig. 6. One reason is that the accuracy of the method has only improved in some specific situations. The CNN-based hybrid model (R7) is accurate in complex backgrounds, yet lower in specific gestures. Conversely, EMI (R21) is precise in each gesture, while degraded in outdoor. Gesture-CNN (R5), I3D (R13), Star RGB (R15) and A two-stream hybrid model (R41) are classified with accuracy, while error rate increased in real-time, issues in a more complex context, losing hand details, and confused finger and hand, respectively. Non-touch character writing system (R16) is accurate without increasing error rates over time. The normalized 2D SDD features and a prior knowledge (R42) accuracy, except for similar hand gestures of classes ‘a’, ‘e’, ‘m’, ‘n’, ‘s’ and ‘t’.

The other reason is that the models for improving accuracy bring extreme consumption costs, which in turn affect the recognition speed, such as R3 (CNN and LSTM-based model), R18 (CNN, multi-scale fusion, weight multi-scale, multi-scale attention and attention U-Net), R25 (F-BiLSTM and F-BiGRU), R29 (RetinaNet-DSC), and R33 (RPCNet). In contrast, the current model for efficiency hardly achieves relatively good accuracy, such as R1 (structured dynamic time warping), R30 (A lightweight 3D Inception-ResNet), and R17 (optimized Darknet CNN).

Many researchers have made efforts to combine various models to obtain a balance, but the outcome is unsatisfactory. An integrated model with the covariance matrix (R11) and the deep single-stage CNN model (R32) balanced the accuracy and efficiency, while they had deficiencies in multiple gestures and in predicting the level of detail, respectively. The hybrid model (R35) balanced computation cost and classification accuracy, yet was still limited on mobile devices.

In addition, although recent methods remarkably contribute to HGR, they still have the following problems in practical applications. Most methods perform excellently on datasets (R13, R21, R28), yet poorly in real-world scenarios, such as complex backgrounds, hydrological change, and dynamic outdoor backgrounds. Moreover, high computational cost is a major bottleneck of model fusion (R3, R13, R18, R25, R26, R29, R33, R47). Although model fusion specialized in accuracy improvement and robustness, they have limited application scenarios on real-time and resource-constrained devices. On the contrary, lightweight models (R13, R38) are more practical in these scenarios, whereas their environmental adaptability is weak. Beyond that, the existing methods, despite improved in some special scenarios, remain to be further advanced, such as, hand-hand or hand-object interaction with occlusion (R2), online (R4, R5), complex backgrounds (R7), lighting variations (R10, R16), underwater (R28), and astronauts interacting with robots (R31).

Current research on HGR has gained remarkable achievements in accuracy, efficiency, and robustness by integrating deep learning models such as CNN, LSTM, attention mechanisms, and multi-modality. However, it still faces challenges such as the balance between accuracy and efficiency, weak environmental adaptability, occlusion, and high consumption cost.

RQ2: what are the performance metrics used to evaluate the HGR models?

In response to RQ2, this section elaborates a statistical summary and critical analysis, focusing on the evaluation metrics, datasets, ablation studies, and technical assessments. Moreover, the comparative analysis and critical synthesis are elaborated in the discussion section. These synthesizing aspect provides valuable insights into performance evaluation and lay the foundation for the future. Meanwhile, these statistical analyses provide valuable insights into the influence on model performance and a crucial foundation for refining dataset design and selection, thereby contributing to enhancing the model’s usability and research significance.

The following statistics have been gathered by reading forty-seven references. The numerical data in the spreadsheet includes evaluation metrics, dataset, ablation study, model performance, and the charts were generated by specialized software.

Evaluation metrics

Figure 8 illustrates the distribution of evaluation metrics in studies. The following is a detailed description.

Figure 8 The analytical evaluation metrics are in two groups, accuracy and efficiency.

The x-axis presents the number of literature that use a certain evaluation parameter.

Accuracy

Commonly evaluating accuracy by recognition accuracy, F1-score, precision, recall, average precision (AP), etc., with almost all models using recognition accuracy to assess the performance. The equations related to the evaluation metrics are listed below:

Precision emphasizes the reliability of positive predictions and indicates the proportion of true positive predictions among all positive predictions:

(1) Precision=TruePositivesTruePositives+FalsePositives.

Recall evaluates the model’s ability to identify all relevant instances, which is vital in minimizing false negatives. It reflects the percentage of true positive predictions among all actual positives:

(2) Recall=TruePositivesTruePositives+FalseNegatives.

F β-score is a metric that adjusts the balance between precision and recall using the parameter β:

(3) Fβ=(1+β2)⋅Precision⋅Recallβ2⋅Precision+Recall.

When β=1, it becomes the F1-score, which gives equal weight to precision and recall. Larger β emphasizes recall, while smaller β emphasizes precision.

Average precision (AP) presents the relationship between precision and recall across different thresholds. It calculates the mean of Precision values at varying levels of recall:

(4) AP=∫01P(r)dr

where r is the recall values ranging from 0 to 1, P(r) is precision as a function of recall r.

Mean average precision (mAP) evaluates multi-class object detection performance. It is the average of AP values across all classes:

(5) mAP=1n∑i=1nAPi

where n is the total number of classes, i is the index of a class (ranging from 1 to n, APi is the Average Precision for class i).

Mean squared error (MSE) is a common evaluation metric that measures the average of the squared differences between the actual values ( yi) and the predicted values ( y^i). n represents the total number of samples. yi is the actual (true) value for the ith observation. ( y^i) is the predicted value for the ith observation.

(6) MSE=1n∑i=1n(yi−y^i)2.

Efficiency

Many studies evaluate efficiency by computational performance, mean detection time, run time, delay, etc. Computational performance is the average per frame processing time. Detection time means time for detecting hand gestures.

In brief, the majority of models concentrate on enhancing the accuracy, as illustrated in Fig. 6. As a result, in Fig. 8, the evaluation metrics for evaluating accuracy are significantly higher than efficiency. For accuracy, recognition accuracy is the largest because it provides a global assessment of the final performance throughout the real-time HGR process. Following is the F1-score, and its relevant precision and recall. On the contrary, other models demonstrate accuracy by evaluating the low error rate. For efficiency, most evaluation metrics are related to time consumption per frame.

Datasets

The number and category of samples in the datasets are significant factors influencing the performance of the model. Table 9 provides a summary of datasets and their corresponding model performance. Figure 9 illustrate the distribution of self-created and public datasets, along with their relationship to model performance, respectively.

Table 9 Datasets and associated performance in literature.

Dataset name	Video or image	Number of samples	Acquisition method	Background	Number of classes	Applied area	Ref. no.: Performance	
Continuous Letter Trajectory (CLT) (Self-created)	Video	1,300	3D camera (Kinect2)	N/A	26	Handwriting trajectories and gesture recognition	R1: 88.8%	
R2 (Self-created)	Sequence	34 sequences with about 20k frames	Two RealSense SR300 sensors to capture hand-object interaction	N/A	34	Hand-object interaction	R2: Mean standard deviation of DoFs is around 0.45	
Jester	Video	148,092	Collected via a crowdsourcing platform	Various	27	Hand gesture recognition	R3: 95.73%, R13: 95.31%, R34: 95.55%	
Nvidia	Video	1,532	The SoftKinetic DS325	Indoor	25	Hand gesture recognition	R3: 85.13% (only selected RGB frames from videos)	
Sheffield Kinect Gesture (SKIG)	Video	1,080 RGB and 1,080 depth videos	Kinect sensor	Uniform	10	Gesture recognition	R4: 99.63%, R10: 93.1%, R11: 98.1%, R12: 100%, R23: 99.7%, R33: 99.70%, R37: 98.11%, R39: 97.87% (RGB) and 98.70% (Depth), R41: 99.6%	
ChaLearn LAP IsoGD	Video	47,933	RGB-D cameras	Various	249	Gesture recognition	R4: 60.27% (RGB) and 57.02% (Depth), R8: 60.2%, R12: 68.14%, R13: 65.13%, R15: 52.18%, R30: 71.37%, R33: 69.3% (RGB) and 69.65% (Depth), R47: 86.1%	
R6 (Self-created)	Image	10,800	A multimodal measurement platform composed of radar and camera	Various	10	Hand gesture recognition	R6: 94.58%	
R10 (Self-created)	Image	4,200	Kinect	Indoor	28	Hand gesture recognition	R10: 92.8% (static), 86.2% (fine-grained), 91.3% (coarse-grained hand gestures)	
EgoGesture	Video, Image	Video: 2,081, Image: 24,161	Intel RealSense SR300	Indoor/Outdoor	83	Egocentric gesture recognition	R4: 93.20% (RGB) and 93.35% (Depth), R20: 93.8%, R21: 90.63%, R33: 93.93% (RGB) and 94.14% (Depth)	
Indian Sign Language (ISL) (Self-created)	Image	2,150	A camera	Indoor	43	Indian sign language recognition	R5: 94.83% (alphabets), 99.96% (static words)	
The JochenTriesch’s	Image	3,000	JochenTriesch’s database	Uniform	10	Sign language recognition	R5: 100%	
DHG-14	Video	2,800	Intel RealSense camera	Indoor	14	Hand gesture recognition	R7: 97.20%, R11: 87.18%, R38: 96.07%, R37: 88.4%, R41: 97.8%	
DHG-28	Video	2,800	Intel RealSense camera	Indoor	28	Hand gesture recognition	R7: 96.3%, R11: 78.61%, R38: 94.4%, R41: 92.1%	
NU-DHGR	Sequence	1,050	Leap motion	Indoor	10	Hand gesture recognition	R11: 98.1%	
FHPA	Video	1,050	Depth camera	Indoor	45	Hand posture recognition	R7: 92.78%	
American Sign Language Lexicon Video (ASLVID)	Video	3,000	Native signers	Indoor	50	American Sign Language recognition	R8: 68.8%, R47: 93%	
RKS-PERSIANSIGH	Video	10,000	Camera	Indoor	100	Persian sign language recognition	R8: 74.6%, R47: 99.5%	
First-Person	Image	105,459 RGB-D frames	Intel RealSense SR300 RGB-D camera	Complex	45	Hand posture recognition	R8: 67.2%, R47: 91%	
SHREC’17 track	Video	2,800	Intel RealSense camera	Indoor	14	3D Hand gesture recognition	R9: 94.17%	
BSG2.0 (Self-created)	Video	10,720	DS325	Indoor	26	Hand gesture recognition	R13: 99%, R30: 98.04%	
Sign language digits	Image	2,062	Webcam	Monochrome	10	Sign language recognition	R14: 98.40%	
Thomas Moeslund’s Gesture Recognition	Image	2,060	Webcam	Monochrome	24	Sign language recognition	R14: 98.09%	
Montalbano	Video	13,206	Kinect 360 sensor	Indoor	20	Gesture recognition	R15: 94.58%	
Gesture Commands for Robot Interaction (GRIT)	Video	543	RGB camera	Indoor	9	Gesture recognition	R15: 98%	
R16 (Self-created)	Image	12,600	Webcam	Monochrome	7	Hand gesture recognition	R16: 98.09% (Offline), 96.23% (real-time) 2.08 s	
R17 (Self-created)	Image	800	Camera	Indoor	4	Human-robot interaction	R17: 96.92%	
OUHANDS	Image	3,000	Intel RealSense F200 camera	Complex	10	Hand shape recognition	R18: 90.9%, R43: 97.49% (0.92 s per frame)	
HGR1	Image	899	Camera	Indoor	27	Hand gesture recognition	R18: 83.8%	
NYU	Video	72,757	Kinect sensor	Indoor	36	Hand pose estimation	R19: Mean error (mm) 8.37	
ICVL	Video	330,000	Intel RealSense camera	Indoor	16	Hand pose estimation	R19: Mean error (AA-A2J 8.37 mm, AA-3DA2J 8.37 mm); FPS (AA-A2J 151.06, AA-3DA2J 79.62)	
MSRA	Video	76,500	Kinect sensor	Indoor	17	Hand pose estimation	R19: Mean error (AA-3DA2J 6.39 mm, AA-A2J 6.30 mm)	
HANDS 2017	Video	957,032	Multiple datasets	Various	32	Hand pose estimation	R19: Mean error (AA-3DA2J 8 mm, AA-A2J 8.08 mm)	
NVGesture	Video	1,532	Webcam	Indoor	25	Hand gesture recognition	R20: 83.2%, R23: 89.8%, R39: 73.03% (RGB) and 85.48% (depth)	
MSR Gesture 3D	Depth sequence	336	Kinect sensor	Indoor	12	Sign language recognition	R21: 99.24%	
Indian Isolated Word Sign (IIWS) (Self-created)	Video	3,000	DSLR camera	Indoor	100	Sign language recognition	R22: 98.75%	
Russian (Self-created)	Image	377,850	DSLR camera	Indoor	1,100	Sign language recognition	R22: 98.75%	
R23 (Self-created)	Video	1,800 RGB and 1,800 depth videos	DS325	Indoor	6	Hand detection and classification	R23: 92.06%	
UTAS7k (Self-created)	Image	7,071	Camera	Indoor	5	Hand gesture recognition	R24: 78.2% (clear), 77.1% (blur)	
Mobile Gesture Database (MGD) (Self-created)	Video	5,547	Inertial sensors	Various	12	Gesture recognition	R25: BiLSTM 98.04%, F-BiGRU 99.15%	
BUAA mobile gesture	Video	1,120	Smartphone	Indoor	8	Gesture recognition	R25: F-BiLSTM 99.06%, F-BiGRU 99.25%	
Smart watch gesture	Video	3,200	Smartwatch	N/A	20	Gesture recognition	R25: F-BiLSTM 95.65%, F-BiGRU 97.40%	
Cambridge hand gesture	Image sequences	900	Camera	Monochrome	9	Hand gesture recognition	R26: 96%, R41: 99.4%	
Sebastien marcel	Image sequences	57	Webcam	Monochrome	4	Hand gesture recognition	R26: 98%	
LeapGestureDB	Video	6,600	Leap Motion	Indoor	11	Hand gesture recognition	R27: 90%	
RIT (Rochester Institute of Technology)	Image	9,600	Leap Motion	Indoor	12	Gesture trajectory	R27: 90%	
R28 (Self-created)	Image	12,106	CADDY Underwater Gestures dataset	Various	16	Diver’s gesture recognition	R28: 85%	
NUSHP-II	Image	2,000	Digital camera	Indoor	10	Hand posture recognition	R29: 99.9%, R32: 99.3%	
Senz-3D	Video	1,320	Creative Senz3D camera	Indoor	11	Hand gesture recognition	R29: 99.99%, R32: 98.2%	
MITI-HD	Image	7,500	Webcam	Indoor	10	Hand gesture recognition	R29: AP 99.21%, AR 96.99%, F1-Score 98.10%, Prediction time 82 ms; R32: 99.6%	
SHRI-VID (Self-created)	Video, Image	Video 102, Image more than 6,000	ASL, SRSSL, Egohands, VIVA Hand Detection datasets	Indoor	3	SHRI robot-astronaut interaction	R31: 91.6%	
AU-AIR	Video	32,823	Drone	Outdoor	8	Object detection	R31: 71.68%	
ImageNet-VID	Video	5,354	Collected from the web	Various	30	Object detection	R31: 64.7%	
R35 (Self-created)	Video	137 videos, 41,100 frames	N/A	Indoor	2	Fingertip tracking	R35: 96.89% in 0.0267 s, 96.82% in less than 0.13 s	
R36 (Self-created)	Image	4,320	Microsoft Kinect	Indoor	36	Hand gesture recognition	Arabic numbers: 95.83% and 97.25%; English alphabets: 91.35% and 92.63%	
26-gesture	Position of dominant-hand forefinger	321	Leap Motion sensor	N/A	14	Gesture trajectory 3D points	R37: 99.3%	
VIVA	Video	885 sequences	Microsoft Kinect	Vehicle (complex background)	19	Hand gesture recognition in vehicles	R39: RGB 81.50%, depth 80.15%	
HA (Self-created)	Image	1,140	Xbox One Kinect, HTC Vive, mouse data	Indoor	17 Kinect, 11 HTC Vive, 10 mouse gestures	Gesture recognition (skeleton and mouse trail)	R40: 95%	
Northwestern University Hand Gesture (NWUHG)	Video	1,050 videos	Motion divergence fields	Indoor	10	Dynamic hand gesture recognition	R41: 98.6%	
American Sign Language (ASL) Finger Spelling Dataset	Image	48,000	Kinect depth camera	Indoor	26	American Sign Language recognition	R42: 89.38%	
Near-infrared (NI) Gesture	Image	2,000	Leap Motion infrared camera	Black	10	Hand gesture recognition	R42: 99%	
The National University of Singapore (NUS)	Image	2,750	The charge coupled device (CCD) cameras	Indoor/outdoor	40	Hand postures	R42: 97.98%	
NITS S-net	Image	5,000	1MP HD webcam	Complex	95 gesture trajectories	Gesture recognition	R43: 99.58%, 0.95 s per frame	
Oxford hand	Image	13,050	Collected from various public datasets	Complex	N/A	Hand detection	R43: 87.01%, 1.15 s per frame	
EgoHands	Image	4,800	Head-mounted cameras	Complex	4	Hand detection	R43: 97.05%, 0.4 s per frame	
Dataset1	Image	25,300	N/A	Monochrome	36	Sign language recognition	R44: 98.29%	
Dataset2	Image	57,000	Collected from other dataset on Kaggle website	Monochrome	27	Sign language recognition	R44: 98.56%	
HG14 (HandGesture14) (Self-created)	Image	14,000	Camera	N/A	14	Hand gesture recognition	R45: 90%	
Fashion-MNIST	Image	70,000	Collected from Zalando’s website	N/A	10	Hand gesture recognition	R45: 93.88%	
CIFAR-10	Image	60,000	Subsets of the 80 million tiny images dataset	Various	10	Hand gesture recognition	R45: 81.42%	
R46 (Self-created)	Image	About 27,495	Microsoft Kinect sensor	Monochrome	26	American Sign Language recognition	R46: 97.4%	

Figure 9 Comparison of the recognition rate between self-created datasets and public datasets.

Researchers train and test models on public and self-created datasets, as illustrated in T, with nineteen studies designed for their datasets. Moreover, some exclusively test and train the model on self-created datasets, some only on public datasets, as well as others on both self-created and public datasets.

In addition, as demonstrated in Fig. 9, on the same range of recognition rates, the percentage of high recognition rates for the self-created datasets is higher than the public datasets.

In brief, the detailed information of datasets and the model performance are summarized in Table 9, a total of seventy-one datasets are scattered and diverse. Many studies design their datasets for training and testing models. The benefit of this is that the model performs better on self-created datasets, as shown in Fig. 9. Some of them choose testing models on not only self-created datasets but also public datasets to demonstrate that theirs is the best. Others select public datasets for training or testing, such as the SKIG dataset (Liu & Shao, 2013), the DHG-14/28 dataset (De Smedt, Wannous & Vandeborre, 2016), and the Jester dataset (Materzynska et al., 2019).

Technical evaluation

The underlying models for the methods are intimately with evaluation metrics. Appropriate evaluation metrics ensure an objective assessment of the effects of method refinement. Meanwhile, the intention of the technical evaluation is also to confirm whether the modified model reaches the intended aim. Table 10, Figs. 9 and 10 demonstrate the distribution of recognition rate, evaluation parameters, and performance.

Table 10 Summary of models and performance.

Ref. No.	Main underlying model	Improvement	Limitation	Performance	Application	
R1	LSTM, Others	Efficiency	The overlap problem, the mismatch problem	CLT (Self-created): F1-Score 88.8%	Handwriting trajectories and gesture recognition	
R2	LSTM, Others	Occlusions between the hand and the object	Geometry ambiguities, severe segmentation errors, no two hands or multiple objects interaction	Self-created dataset: mean standard deviation of DoFs is around 0.45	Hand-object interaction	
R3	CNN, LSTM	Accuracy	Computing cost	Jester: 95.73%; Nvidia: 85.13%	HGR	
R4	CNN, LSTM	Accuracy	Challenge in online gesture recognition	SKIG: 99.63%; ChaLearn LAP IsoGD: 60.27% RGB, 57.02% Depth; EgoGesture: 93.20% RGB, 93.35% Depth	HGR	
R5	CNN	Classification accuracy	Error rate in real-time	ISL (Self-created): alphabets 94.83%, static words 99.96%; JochenTriesch’s: 100%	Sign language recognition	
R6	CNN, LSTM, Others	Robustness; Generalization ability for classification tasks in diverse FOV scenes	Consumption cost	Self-created dataset: 94.58%	HGR	
R7	CNN	Accuracy in a complex background	Low accuracy in a specific gesture	DHG-14: 97.20%, DHG-28: 96.3%; FHPA: 92.78%	3D HGR	
R8	CNN, LSTM, Attention Mechanism	Accuracy	Zero-shot learning	74.6%, 67.2%, 68.8%, 60.2% on RKS-PERSIANSIGN, First-Person, ASLVID, ChaLearn LAP IsoGD respectively	Sign language recognition, hand posture recognition	
R9	Attention mechanism	Accuracy, lower computational complexity	Occlusion	SHREC’17: 94.17%	3D HGR	
R10	CNN	Insensitivity to changes in light conditions	Trajectory differentiation of multiple hands in the same region	SKIG: 93.1%; Self-created: static 92.8%, fine-grained 86.2%, coarse-grained 91.3%	HGR	
R11	CNN, Attention mechanism	No need for GPUs, accuracy, efficiency	No multi-hand gesture recognition	NU-DHGR: 98.10%; SKIG: 98.10%; DHG-14: fine 76.10%, coarse 93.44%, all 87.18%; DHG-28: 78.61%	3D HGR	
R12	CNN	Accuracy	Influence of gesture-irrelevant factors	ChaLearn LAP IsoGD: 68.14%; SKIG: 100%	HGR	
R13	CNN	Robust classification, faster response, smaller storage	Issues in more complex context	ChaLearn LAP IsoGD: 65.13%; Jester: 95.3%; BSG2.0: 99%	HGR	
R14	CNN	Accuracy	Robustness	Sign language digits: 98.40%; Thomas Moeslund’s: 98.09%	Sign language recognition	
R15	CNN	Classification accuracy	Not suitable for moving camera; losing hand details	Montalbano: 94.58%, GRIT: 98%, ChaLearn LAP IsoGD: 52.18%	HGR	
R16	CNN	Accuracy, Robustness	Increasing error rates over time	Self-created dataset: Offline 98.09%, Real-time 96.23%, Error 2.07%	HGR	
R17	CNN, Attention mechanism	Efficiency	Only for a small number of classes	Self-created dataset: 96.92%, Detection time 0.1461 s	Human-robot interaction	
R18	CNN, Attention mechanism	Accuracy	Computational cost for mobile devices	OUHANDS: 90.9%, HGR1: 83.8%	HGR	
R19	CNN, Attention mechanism	Reducing error rate, improving runtime	Evaluated only on NYU, not others	NYU: Mean error AA-A2J 8.37 mm; FPS: AA-A2J 151.06, AA-3DA2J 79.62; ICVL: 6.30 mm; MSRA: 8.08 mm; HANDS 2017: 8.27 mm	Hand pose estimation	
R20	CNN, Attention mechanism	Reducing computation costs, improving accuracy	Attention-based model uncertainty	NVGesture: 83.2%; EgoGesture: 93.8%	HGR	
R21	CNN, LSTM, Attention, others	Recognition accuracy and precision	Degraded in outdoor environments	EgoGesture: 90.63%; MSR Gesture 3D: 99.24%	HGR, sign language recognition	
R22	Others	Accuracy	No continuous sentences	Self-created dataset: 98.75%	Sign language recognition	
R23	CNN, LSTM	Zero/negative delay, efficiency	Failing on embedded platforms	Self-created: 92.06%; NVGesture: 89.8%; SKIG: 99.7%	Hand detection and classification	
R24	CNN, Others	Robustness	Accuracy	Self-created: 78.2% (clear), 77.1% (blur)	HGR	
R25	CNN, LSTM	Accuracy	Computational time	MGD: BiLSTM 98.04%, F-BiGRU 99.15%; BUAA: 99.25%; SmartWatch: 97.40%	Mobile HGR	
R26	CNN, LSTM, Attention	Robustness	Computational efficacy not quantified	Cambridge Hand Gesture: 96%; Sebastien Marcel: 98%	HGR	
R27	CNN, LSTM	Efficiently classifying	More time-consuming	LeapGestureDB, RIT: 90%	HGR	
R28	CNN, LSTM, Attention	Accuracy	Duplicate recognition, low mAP, poor segmentation in water	Self-created: 85%	Diver’s gesture recognition	
R29	CNN	Precision	Computation time	NUSHP-II, Senz-3D, MITI-HD: 99.1–99.99%	HGR	
R30	CNN, Others	Efficient feature extraction, low storage	Recognition accuracy	Self-created: 98.04%; IsoGD: 71.37%; 762 fps GPU, 37 fps CPU	HGR	
R31	CNN, LSTM, Attention, Others	Detection accuracy and speed of small objects	Cannot distinguish left/right hand	SHRI-VID: 91.6%; ImageNetVID: 64.7%; AU-AIR: 71.68%, 23fps	Robot-astronauts interaction	
R32	CNN	Time-efficient, high precision	Inaccurate at detail level	MITI, Senz-3D, NUSHP-IIs: 98.2–99.6%	HGR	
R33	CNN, Attention mechanism	Accuracy	Delay, error detection in HCI	EgoGestures: RGB 93.93%, Depth 94.14%; SKIG: 99.70%; IsoGD: 69%	HGR	
R34	CNN, LSTM	Accuracy, reduce misrecognition	Asymmetric gesture mapping	Jester: 95.55%, Time: 17.61 s	HGR	
R35	LSTM, Others	Low computation, accuracy	Limited to Android/Apple	Self-created: 96.89% in 0.0267 s	Mobile HGR	
R36	Others	Accuracy	Right-handers only tested	Self-created: Arabic numbers: 96%; English alphabets: 92%	3D trajectory	
R37	CNN	Accuracy	Robustness	SKIG: 98.11%; DHG2016: 88.4%; 26-gestures: 99.3%	HGR	
R38	LSTM, Others	Robust, low cost	Spatio-temporal feature diversity not integrated	DHG-14/28: 96.07%, 94.4%	HGR	
R39	CNN, Attention mechanism	Accuracy	Only unimodal RGB performs well	SKIG: RGB 97.87%, Depth 98.70%; VIVA: RGB 81.5%, Depth 80.15%; NVGesture: RGB 73.03%, Depth 85.48%	3D HGR	
R40	Others	Efficiency, precision	Not orientation invariant; endpoint issues	HA (Self-created): 95%	HGR	
R41	CNN, LSTM, Others	Classification accuracy	Whole hand vs. finger confusion	NWUHG: 98.6%; SKIG: 99.6%; Cambridge: 99.4%; DHG14/28: 97.8%, 92.1%	HGR	
R42	CNN	Accuracy	Lower accuracy on similar gestures	NUS II subset A: 97.98%; ASL: 89.38%; NI: 99%	Sign language recognition	
R43	LSTM, Attention, Others	Reduce compute time, blurring	Noise in trajectory	Multiple datasets: all above 96–99% across bare hand detection and SDT	Trajectory	
R44	CNN, LSTM	User-friendly, intuitive	Rotation, scale, translation problems	Dataset1: 98.29%, Runtime 1.227 s; Dataset2: 98.56%, Runtime 1.1136 s	HCI	
R45	CNN, LSTM	Accuracy	No regularization	HG14: 90%; FashionMnist: 93.88%; Cifar-10: 81.42%	HGR	
R46	CNN, Attention mechanism	Lighting condition robustness	Non-outdoor conditions	Self-created: 97.3%, Error 2.6%, Recognition time 0.013 s	HGR	
R47	CNN, LSTM	Accuracy, efficiency	No complex signs included	RKS-PERSIAN SIGN: 99.5%; First-Person: 91%; ASVID: 93%; IsoGD: 86.1%	Sign language recognition	

Figure 10 Recognition rate distribution of key stages and underlying models.

In brief, in high recognition rate areas, the biggest ones are feature extraction and classification. It is followed by CNN and model combinations, as shown in Fig. 10. From another perspective, many studies refined the performance of feature extraction to improve the integration accuracy. Meanwhile, it further reflects that CNN contributes the most to enhancing the accuracy. Furthermore, methods with CNN and multiple underlying models tend to select diversity evaluation parameters.

Ablation study

In the deep learning field, an ablation study is a significant experimental method. It is employed to understand and evaluate the contribution of each component in a model to its overall performance. Among the forty-seven articles, nine chose ablation experiments, illustrated in Fig. 11. The details include: Similar underlying models: R8, R12, R18, R21, R31, R39. Comparing the enhanced underlying model to similar ones demonstrates that the improved model generates the greatest contribution overall.

Integrating specific models: R2, R9, R12, R19, R20, R31. Evaluating the overall efficacy of integrating a certain underlying model.

Pre-processing: R8, R9, R12. Testing the effectiveness of pre-processing techniques such as single or multiple data inputs and data augmentation on overall performance.

Others: Sense (R6); quantity of sub-videos (R21); background and scale suppression, frame-to-frame temporal consistency (R31); adaptive parameter (R39)

Figure 11 Compare the recognition rate with and without ablation study.

Furthermore, the recognition rates with and without the ablation study are compared for awareness of the effect of the ablation study, in Fig. 11.

In brief, fewer researchers completed ablation experiments. Most studies contrast the updated models to those of similar underlying models. The majority of them are multi-model fusion. In addition, analyzing these studies revealed that whether or not to do ablation experiments has little general impact, as shown in Fig. 11.

Comparative evaluation across datasets

This section provides a comprehensive analysis of the impact of dataset attributes, such as size or number of classes on model performance, comparing the performance of various techniques on different datasets.

Impact of datasets attributes on model performance

Seventy-one datasets were discovered in the forty-seven literature. These datasets vary in size, ranging from small, such as Sebastian Marcel with 57 samples, to huge ones like ImageNet-VID with a million samples. Overfitting is a problem for small datasets, while large datasets require increased computational resources and training time.

Moreover, both the number of samples and category diversity of datasets can directly influence the performance of models. The accuracy of some models decreases as the number of samples in the dataset increases (R42). Model performance in R42 on NI with 2,000 samples, NUS with 2,700 samples, and ASL with 48,000 samples achieves 99%, 97.98%, and 89.38% accuracy, respectively. Moreover, the number of samples and categories in datasets can directly influence the performance of models. The accuracy of some models decreases as the number of samples in the dataset increases (R42). Model performance in R42 achieves 99%, 97.98%, and 89.38% on NI with 2,000 samples, NUS with 2,700 samples, and ASL with 48,000 samples, respectively. Meanwhile, the accuracy of some models improves with the number of categories reduced (R4, R5, R12, R13, R30, R31, R33). Many models can achieve accuracy over 90% on small datasets (e.g., R4: 99.63% on SKIG, R12: 100% on SKIG, R15: 98% on GRIT). However, their performances drop to around 50–80% on large and diverse datasets like ChaLearn LAP IsoGD (e.g., R4: 60.27% (RGB) and 57.02% (depth), R12: 68.14%, R15: 52.18%) due to increased intra-class variability and inter-class similarity. Similarly, model performance in R31 improves significantly as the number of categories in the dataset decreases. R31 on ImageNet-VID with 30 categories, AU-AIR with eight categories, and SHRI-VID with three categories achieves 64.7%, 71.68%, and 91.6% accuracy, respectively. This is because models trained on datasets with numerous categories inherently have elevated generalization demands, weakening their accuracy.

Furthermore, datasets with skewed class distributions make it difficult to achieve balanced learning and reliable model evaluation. For example, Indian sign language (ISL) (self-created) contains only 2,150 images spread across 43 categories (R5). HGR1 contains only 899 images spread across 27 categories (R18).

In addition, some models are simply selected for training on small, less classified datasets, and the generalization ability of their models needs to be further tested. For instance, R26 (Hough transform and artificial neural network) obtains the accuracy of 96% on the Cambridge Hand Gesture with 900 samples and 98% on Sebastien Marcel with 57 samples.

Technical evaluation across datasets

This section evaluates the models from two aspects. One is the models compared and evaluated on the same dataset, which provides a fair representation of the effect of various techniques on model performance. This is because there is no dataset heterogeneity in the same dataset, facilitating a straightforward comparison of the effects of algorithm design, strategy optimization, and technical architecture. Another aspect is the analysis of the generalization ability of the model on different datasets, because a single dataset is specific and cannot be examined for the generalization of the model.

On the same dataset, most of the models have a mirror difference in performance, but there are some exceptions. Comparing the methods on ChaLearn LAP IsoGD, the model in R47 using CNN-based models with a mathematical technique is far more accurate than the others by about 20%. Models with hybrid technologies in R8 and R47 are implemented on the same datasets, yet their accuracy is off by more than 20%. Moreover, the integrated framework based on the covariance matrix in R11 is lower than other methods by around 10% on DHG-14/28. In addition, the performance of models trained, validated and tested on SKIG and EgoGesture is excellent.

Evaluating the generalization ability of models involves two parts: cross-dataset, multiple classes. Many models have good performance on different datasets. RetinaNet-DSC in R29 and Hybrid-SSR in R32 are both CNN-based models, achieving nearly 100% accuracy on three datasets. The model in R47 utilized CNN-based models with a mathematical technique, obtaining high accuracy on the other three datasets. Moreover, some models obtain high accuracy on multiple-class datasets. The hDNN-SLR framework in R22 achieves 98.75% on the Russian dataset with 1,100 categories, indicating it can handle complex data.

RQ3: which research gap remains in real-time HGR using deep learning?

Real-time HGR requires instant response and recognition accuracy. Although deep learning improves some accuracy and efficiency issues, challenges remain ongoing.

In this section, data was created by reading the introduction of each selected literature and analysis from the previous two research questions. The statistics summarized in the spreadsheet are used to generate a pie chart.

Challenges of existing methods

Challenges are still present for real-time HGR. From the details of limitations in Table 10 the main issues are centered on accuracy in some situations, such as mismatch, ambiguities, and occlusion. The following are limited applications, low efficiency, poor environmental adaptation and high consumption cost. These limitations related to extant research gaps exist primarily in acquisition devices, environmental complexity, the effects arising from hand movements, and the disadvantages of models themselves.

Accuracy

The unsatisfactory accuracy issue accounts for the highest percentage of all limitations, such as Acquisition device: The camera quality impacts the performance of the HGR model (Dubey, 2023). It is a little difficult to obtain multi-modal data (Li et al., 2021a).

Feature extraction: Diversity handcrafts features are not combined into spatiotemporal feature (Do et al., 2020). Mismatch and overlap problems normally happen in the trajectory. Recognition errors because of the similarity of hand poses always existed in this situation (Xiao et al., 2023). Hands move or rotate fast in a real-time system. The occlusion and Geometry ambiguities from self-similar hand and fingers in hand-object interaction (Zhang et al., 2019).

Models: The differentiated hand movements reversed direction (Wang et al., 2023a) due to some models.

Application range

Although application is the ultimate goal of designing models, many models have some limitations in their applications. For gesture recognition, some were limited or failed in online (Tang et al., 2021), embedded platforms (Lu et al., 2024), Android and Apple systems (Hou et al., 2023). For the testing part, they tested with no multi-hand (Fang et al., 2019), only right-handers (Liu et al., 2019), a small number of classes (Tellaeche Iglesias et al., 2021), and simple sign samples (Rastgoo, Kiani & Escalera, 2022). For trajectory, it was difficult to distinguish the differentiation of multiple hands in the same region (Zhang, Tian & Zhou, 2018).

Efficiency

Most inefficiency problems are generated by the design of the method or the model itself. For bidirectional long short-term memory (BiLSTM) and bidirectional gated recurrent unit (BiGRU), the computational burden is increased due to more neurons representing bidirectional memory (Li et al., 2018). The HBU-LSTMa model combined the BiLSTM and the U-LSTM (Ameur, Khalifa & Bouhlel, 2020), and the optical flow algorithm (Zhang, Wang & Lan, 2020) is very costly.

Environment adaptation

The hand is recognized inaccurately in complex environments including changes in lighting conditions, partial occlusions, and color variations (Tellaeche Iglesias et al., 2021; Yadav et al., 2022; Dubey, 2023; Jiang et al., 2021; Jain, Karsh & Barbhuiya, 2022; Haroon et al., 2022; Jingchun, Su & Sunar, 2024).

Consumption cost

The size of the input data influences the recognition performance. High resolution enhances recognition performance, yet it consumes memory. Thus, testing at the edge with memory limits while maintaining classification accuracy (Liu & Liu, 2023) is required.

Datasets

The dataset is fragmented, as shown in Table 9, with a lack of standardization and weak generalization of the model. Many existing categorical datasets have significant deficiencies in terms of variety, quantity and data imbalance, failing to supply the extensive and diverse data required by deep learning models. Hence, this limits the generalization ability.

Imbalance between accuracy and efficiency

Many researchers have endeavored to balance accuracy and efficiency, yet the results are only accomplished at the cost of some partial accuracy.

Future research direction

Based on the analysis of research gaps, potential research directions, but not limited to, are summarized into the following aspects: Spatiotemporal feature extraction. In the context of real-time HGR, the continuity and precision of spatiotemporal features play a pivotal role in addressing challenges associated with hand movements. These include issues such as motion blur, occlusion, and overlapping gestures caused by rapid hand movement. Researchers can explore innovative techniques for spatiotemporal feature extraction to improve the robustness and accuracy of HGR systems.

Extending this exploration, efforts can focus on designing a multi-path intersection mechanism with a cross-scale attention fusion module. One path is dedicated to fast finger movements, another captures slow palm and wrist location, and other paths predict hand postures by applying principles of dynamics. Meanwhile, the cross-scale attention is applied to extract local micro-moving and global gesture dynamics.

Model fusion for multi-modal. Acknowledging that individual models may have inherent limitations, researchers can investigate the integration of multiple models to achieve complementary functionalities. This approach can lead to a significant enhancement in system performance by leveraging the strengths of different models. Developing methodologies for efficient model fusion can be a promising avenue for advancing technology with multi-modal.

Following this line of inquiry, researchers can channel their efforts into enhancing multi-modal fusion by incorporating dynamic sensor calibration techniques. Specifically, develop a cross-sensor alignment algorithm using unsupervised learning to automatically calibrate visual, depth, and inertial data in real-time. Implement a hierarchical fusion strategy where low-level motion cues from inertial sensors refine high-level visual features, improving recognition precision for subtle or occluded gestures.

A lightweight and efficient neural network. With the widespread adoption of mobile devices for various applications, ensuring a seamless user experience is paramount. Researchers should focus on developing lightweight and efficient neural network architectures for mobile devices. This endeavor aims to enable users to operate machines and interact with digital interfaces effortlessly, particularly through mobile terminals. Investigating efficient model architectures for mobile deployment can pave the way for user-centric HGR applications.

Subsequent initiatives can address developing ultra-efficient neural networks by integrating adaptive sparsity and dynamic pruning during both training and inference. Specifically, the structured pruning fused with reinforcement learning to remove redundant neuron connections in real-time HGR without compromising accuracy. Further, a multi-scale feature compression module is implemented to maintain spatial resolution to minimise computational cost on resource-constrained mobile devices.

Another idea is to develop energy-efficient gesture recognition systems through model quantization and asynchronous execution. Implementing post-training quantization with mixed-precision techniques reduces the memory overhead while retaining performance. Meanwhile, an asynchronous execution framework is designed to process low-priority tasks (e.g., background updates) in parallel to ensure identifying the critical gestures.

Seamless integration with emerging technologies. The rapid evolution of technology unlocks potential opportunities for HGR applications in conjunction with emerging technologies, such as AR, VR, and Internet of Things (IoT). Researchers can explore the seamless integration of HGR with these technologies to create innovative and immersive user experiences. Consider the potential for cross-disciplinary collaborations and holistic system design to unlock the full potential of HGR in these domains.

Continuing along this line of study, researchers can focus their efforts on developing context-aware domain generalization techniques to handle environmental variations, such as complex backgrounds. Meta-learning is introduced to train models to adapt in unexplored scenarios by learning environment-specific parameters. Moreover, comparative learning is utilized to capture cross-environment invariant domains, ensuring robust performance in indoor, outdoor and dynamic scenes.

Normalization and consolidation of datasets. Data quantity, diversity, and numerical balance of each category are core issues to be addressed in datasets. However, existing HGR datasets are scattered, cluttered, and lack standardization. Expanding upon this research focus, researchers can standardize and integrate similar sign language recognition datasets while supplementing the missing gesture categories. Simultaneously, the gestures are updated regularly following the rhythm of language updates. Moreover, the researchers can apply various acquisition devices from multiple viewpoints to capture diverse hand features. This can support the deep learning model to learn rich features to improve model performance.

Synthesis of findings

Based on the statistical and categorical analysis of the aforementioned data, we summarized the main findings. As Fig. 12 presents, model performance depends on the interconnection of technical aspects, evaluation metrics, datasets, ablation studies, etc.

Figure 12 The main findings.

On the technical dimension, model improvement relies on technological architecture design and data pre-processing enhancement. CNN-based models contribute significant strengths in improving accuracy. Even though multi-model fusion has become a mainstream method, its effect is inconsistent. Much research focuses on enhancing feature extraction techniques, while pre-processing is comparatively minor. Nevertheless, data cleaning, denoising and data augmentation are significant in refining the accuracy of models. Meanwhile, pre-processing is crucial for solving the imbalance problem in small-sample datasets, which can effectively enhance the stability and predictive ability of the model.

Moreover, the datasets are dispersed, seventy-one datasets involved in forty-seven literature, and these datasets demonstrate uneven differences. Meanwhile, dispersed datasets also indicate the limited uniformity level of some research fields. Compared to the total number of samples, the number of categories holds considerable weight on model performance. The models ordinarily yielded analogous performance under the same dataset. In comparison, the performance disparity between the models is marked under multi-category and large datasets. In particular, the self-created dataset exhibits uniqueness, created for special studies. The referential value of model performance on self-created large-scale and multi-category datasets is significantly higher than that of small-scale datasets. Models that perform exceptionally on several large multi-category datasets generally have superior generalization and robustness. In particular, ablation studies are critical for researchers to understand the impact of model structure, especially in model fusion. However, few studies have conducted ablation experiments in the literature.

In addition, the original aim of designing the HGR models is to facilitate widespread applications. However, the reality is that existing models have weak generalization abilities caused by dataset heterogeneity. Generalization ability refers to the performance of models on unseen data. Dispersed datasets, which imply increased disparities among samples, result in weak generalization efficacy due to challenges for models in capturing uniform and sufficient features during training. Meanwhile, despite some studies emphasising the robustness of their models, discrepancies exist between the results and the performance across datasets. Few models have been rigorously tested in a real-world scenario.

In brief, improving a model requires a multidimensional method, considering underlying models, module functionality, model structure, dataset attributes, evaluation metrics, application scenarios, etc.

Discussion

In this review, a comprehensive exploration of vision-based deep learning methods and evaluation techniques in the domain of real-time HGR from 2018 to 2024 has been undertaken. The investigation, structured around the model, model evaluation, and research gaps, has not only uncovered existing trends but has also illuminated the way forward.

The predominant inclination among current HGR models towards model fusion and self-created datasets has yielded valuable insights. Single models, each with its inherent strengths and limitations, have demonstrated the power of precision. In contrast, model fusion, while enhancing performance, has posed the challenges of model size and resource consumption. It’s a trade-off that must be navigated. Moreover, the paramount role of carefully chosen datasets in shaping model performance cannot be overstated, highlighting the need for versatile, high-quality data sources to propel the field forward.

Current datasets are disorganized with remarkable divergence in size and number of typologies. Simultaneously, insufficient data support is still lacking in some specific fields. This situation brings significant challenges to model training and evaluation. Optimization of structure and coverage of the dataset significantly contributes to improving the generalization of models, mitigating the effects of inadequate data in specific domains, and providing more reliable data support for subsequent research.

Appraisal models and ablation studies are inescapable for quantifying holistic model capabilities. The composition of the evaluation matrix is peculiarly salient. The selection and analysis of evaluation parameters are indispensable in scientific, comprehensive, reasonable, and without druthers. Ablation study is the cornerstone for researchers to deep-dive into model relevance and composition. It is also an optimal study for model fusion. However, current HGR models have rarely done this study.

HGR models have made significant strides, prompting researchers to explore the balance between high accuracy and computational efficiency. The powerful feature extraction capabilities of CNNs have excelled in accuracy, while LSTMs are widely used in continuous gesture recognition due to their advantages in capturing temporal features. However, their reliance on a large amount of training data also increases the demand for computational resources. Emerging models like Transformers and the attention mechanism are gradually transforming the field. Transformers with their self-attention are particularly effective in extracting essential features while maintaining efficiency in large-scale and diverse datasets. The current research trend is to focus on reducing computational costs while enhancing accuracy, driving the development of hybrid architectures that integrate CNNs, LSTMs, and attention mechanisms, etc.

Yet, beyond these challenges lie immense opportunities. The identified research gaps, including inaccuracy, inefficiency, limited application range, and environmental adaptability, form the foundations for future innovation. The answer lies in the development of seamlessly integrated models, leveraging state-of-the-art technologies, promising not only to overcome these challenges but to extend HGR into uncharted territories. Imagine applications in AR gaming, where HGR can revolutionize user experiences on mobile devices.

Conclusion

This review provides an in-depth and comprehensive analysis of forty-seven studies in three research questions and finds that model performance is intrinsically linked to the underlying model, technical features, model composition, and dataset. CNN-based models are highly accurate, while multi-model fusion is inconsistent. Models perform excellently on self-created datasets. The number of categories in the dataset affects performance, outweighing their size. Multi-category cross-datasets appraise model generalization and robustness. Other major findings include: more research on feature extraction, less application of pre-processing, dispersed datasets, more self-created datasets, fewer ablation experiments, and weak generalization ability and robustness. These findings inform future research directions, such as model enhancement, short-term efforts focus on hybrid underlying models for enhanced accuracy and efficiency, while long-term goals aim to develop multi-modal AR/VR systems to improve immersive interaction. Therefore, this review is not just a collection of insights; it is a road map for progress. As researchers and practitioners, it is our collective responsibility to embark on this journey. This work is envisioned as a catalyst for your future projects, an inspiration to push the boundaries of HGR, and a guide to navigate the intricate landscape of hand gesture recognition. It is imperative to recognize the limitations of this review, particularly its scope and depth. In future endeavors, the pledge is to expand horizons to analyze the pivotal technologies, datasets, and evaluation metrics that have shaped the history of real-time HGR. This expansion will enable a more comprehensive contribution to this dynamic field. In conclusion, the exciting era in hand gesture recognition stands at a crossroads. Challenges have been laid bare, and the path to progress illuminated. Let this review serve as your compass, propelling all to craft innovative solutions and to steer real-time HGR towards a future where the possibilities are boundless.

Supplemental Information

Supplemental Information 1 Supplemental Figures.

Additional Information and Declarations

Competing Interests

The authors declare that they have no competing interests.

Author Contributions

Cui Cui conceived and designed the experiments, performed the experiments, analyzed the data, performed the computation work, prepared figures and/or tables, reviewed drafts of the article, and approved the final draft.

Mohd Shahrizal Sunar conceived and designed the experiments, reviewed drafts of the article, supervision, and approved the final draft.

Goh Eg Su reviewed drafts of the article, and approved the final draft.

Data Availability

The following information was supplied regarding data availability:

This is a literature review.

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
