# Peer review of "Deep vision-based real-time hand gesture recognition: a review"

_PeerJ Computer Science, doi:10.7717/peerj-cs.2921_

## Round 0.1 · original submission · Major Revisions

The reviewers have recognized the relevance of your work, but they have raised several concerns that must be addressed before the manuscript can be considered for publication. In light of their feedback, we are requesting a major revision, during which all reviewer comments should be carefully incorporated. Significant revisions or extensions are required across various sections to improve soundness and clarity, from the abstract to the conclusions and future work. Please ensure that all comments are fully addressed.

·

Basic reporting

No comment

Experimental design

No comment

Validity of the findings

In this paper, the authors explore vision-based deep learning methods and evaluate multiple techniques in the domain of real-time Hand Gesture Recognition.
The paper is well-structured and elaborative. The related review of the existing related work is well described in Table 1. Page 8 lists the dataset details and evaluation metric rather briefly and is missing essential key information about the size of the dataset. A detailed summary of the datasets and evaluation metrics is necessary and currently missing.
The three research questions answered by the paper are well presented. The size of Figure 6 should be relative to the other figures and currently stands out. The results presented in Figure 10 are unclear, with no legend for the x-axis.

Cite this review as

Reviewer 2 ·

Basic reporting

--In the abstract, you write as follows: "The choice is critical since different tasks require different evaluation parameters, and the model learns more patterns and features from diverse data." Which choice are you referring to?
--In the abstract, you claim that the ablation study is summarized in four aspects. Be explicit on these four aspects.
-- Rewrite the INTRODUCTION section so that the problem domain is very clear to the readers.
-- Your first contribution reads as follows: "To summarize and discuss the last seven years of real time hand gesture recognition methods using deep learning." You have to explain the rationale for choosing seven years in this study.
--- In your second contribution, you sought to analyze and discuss the evaluation metrics and datasets for evaluating, training, and testing the model. Be explicit on the model that you insinuate here.
---You indicated that this is a review paper. However, you have not presented a systemic synthesis of the past research works. It was expected that you would use tables and diagrams to present the major findings from the surveyed literature. However, this has not been done.
-- You have presented some numerical results, but you have failed to vividly describe the procedures adopted to obtain this data.
-- The conclusion section should be written as one continuous paragraph. You need to articulate the main findings of this study based on the study objectives.

Experimental design

--You need to rewrite the methodology section so that it is clear to the readers.

Validity of the findings

--The authors have presented some numerical data. However, it is not clear how these empirical values were arrived at. Therefore, these results lacks validity.

Additional comments

--The structure of this paper needs to be relooked at as it seems inappropriate.
-- Be consistent with the font type and font size. For instance, the font size in Table 3,Figure 9, Figure 11, seems different from the rest of the document.

Cite this review as

Reviewer 3 ·

Basic reporting

1. How is hand gesture recognition used in human-computer interaction (HCI) and related fields?
2. The abstract jumps from introducing deep learning models to evaluation metrics without a clear transition. Use a transition sentence to connect the discussion of models with evaluation metrics, datasets, and ablation studies.
3. The phrase "The ablation study is summarized in four aspects." is vague. Specify what these four aspects are for clarity.
4. Some references are from 7 years ago, while others are from 6 years ago. To maintain consistency, consider standardizing the time frame for the reviewed literature.

Experimental design

Lack of Critical Analysis and Detail: One of the primary concerns with the paper is the insufficient critical analysis of the referenced works. The paper mentions various techniques but lacks a deep evaluation of their strengths, weaknesses, and trade-offs. A more comprehensive analysis of the referenced studies, with a focus on their contributions, limitations, and performance in practical scenarios, is essential for a review paper. The absence of such analysis reduces the paper's overall impact.
Dataset and Preprocessing Gaps: The discussion on datasets is inadequate, as it fails to provide essential details on the challenges associated with the datasets, the diversity of the samples, and how they impact model performance. Furthermore, the section on preprocessing lacks clarity on the necessity of specific techniques and how they contribute to improving model accuracy and robustness. These gaps need to be addressed in order to provide a clearer understanding of the methodology.
Comparative Evaluation of Methods: The paper does not include a detailed comparative evaluation of the methods discussed. A comparison of the techniques' performance, especially across different datasets or environmental conditions, would help readers better understand their effectiveness. This comparative analysis is crucial for the readers to determine the most suitable techniques for their tasks.

Validity of the findings

Future Scope and Research Directions: While the authors have mentioned some broad future directions, the discussion is too general and lacks specificity. The future research directions need to be more concrete, identifying specific challenges or areas where current methods fall short, and suggesting detailed, actionable improvements.

Additional comments

Grammatical and Structural Issues: The paper also suffers from several grammatical issues and unclear sentence structures, which hinder readability and reduce the overall clarity of the paper. These issues need to be addressed for a more polished presentation of the ideas.

Cite this review as

---

## Round 0.2 · Minor Revisions

The review is comprehensive and well-structured, providing interesting figures and charts. The authors have addressed all the reviewers' comments, and from a content perspective, the paper is in good shape.

However, there are still some readability and language-related issues that need attention. The paper generally conveys technical content clearly but it requires some editing before it is ready for publication.

Some examples include:
• Missing or incorrect use of commas, periods, and spaces after punctuation.
• Inconsistent use of past and present tenses when describing similar actions.
• Several English language errors that need correction.
• Some tables are difficult to read due to their dense structure or content.
• Sections with numerous references to other papers can be hard to follow.
• A few repetitive phrases and ideas could be eliminated.

For these reasons, I recommend a minor revision to address the issues reported above. Please consider having the paper proofread by a fluent English speaker or using tools to improve the language.

**Language Note:** The Academic Editor has identified that the English language must be improved. PeerJ can provide language editing services - please contact us at [email protected] for pricing (be sure to provide your manuscript number and title). Alternatively, you should make your own arrangements to improve the language quality and provide details in your response letter. – PeerJ Staff

Reviewer 2 ·

Basic reporting

The authors have addressed all the previous comments and hence the paper can be accepted.

Experimental design

No comment

Validity of the findings

No comment

Additional comments

No comment.

Cite this review as

---

## Round 0.3 · accepted · Accept

The authors have addressed all the comments. The manuscript now reads more fluently.